# Virus-Induced Gene Editing and Its Applications in Plants

**DOI:** 10.3390/ijms231810202

**Published:** 2022-09-06

**Authors:** Chao Zhang, Shanhe Liu, Xuan Li, Ruixuan Zhang, Jun Li

**Affiliations:** State Key Laboratory of North China Crop Improvement and Regulation, College of Life Sciences, Hebei Agricultural University, Baoding 071001, China

**Keywords:** genome editing, CRISPR/Cas9, virus, delivery, Geminivirus

## Abstract

CRISPR/Cas-based genome editing technologies, which allow the precise manipulation of plant genomes, have revolutionized plant science and enabled the creation of germplasms with beneficial traits. In order to apply these technologies, CRISPR/Cas reagents must be delivered into plant cells; however, this is limited by tissue culture challenges. Recently, viral vectors have been used to deliver CRISPR/Cas reagents into plant cells. Virus-induced genome editing (VIGE) has emerged as a powerful method with several advantages, including high editing efficiency and a simplified process for generating gene-edited DNA-free plants. Here, we briefly describe CRISPR/Cas-based genome editing. We then focus on VIGE systems and the types of viruses used currently for CRISPR/Cas9 cassette delivery and genome editing. We also highlight recent applications of and advances in VIGE in plants. Finally, we discuss the challenges and potential for VIGE in plants.

## 1. Introduction

The clustered regularly interspaced short palindromic repeats (CRISPR)/CRISPR-associated protein (Cas) system is an adaptive immune mechanism against invading nucleic acids in archaea and bacteria [1]. Based on their use of DNA–RNA recognition and sequence-specific cleavage, CRISPR/Cas systems have been successfully developed as programmable RNA-guided endonucleases for genome editing since 2012 [2]. After a decade of development, CRISPR/Cas technology has become the most widely used gene editing tool in gene function research, gene therapy, and molecular breeding for crop improvement [3,4]. CRISPR/Cas technologies enable the precise and efficient genetic manipulation of various crop species; thus, they may be applied to crop improvement, agricultural breeding, and wild species domestication [5,6,7]. The CRISPR/Cas9 system can be used to create insertions or deletions (indels) in the coding region of target genes, leading to gene knock-out [8,9,10]. Multiplexed single-guide RNAs (sgRNAs) within a single construct allow for the multiplex editing of one gene or multiple genes simultaneously [10,11]. The fusion of a catalytically deficient Cas9 to different effectors allows for epigenomic editing and gene regulation [12,13]. Furthermore, newly developed technologies with increased precision (e.g., base editing and prime editing) have enabled the creation of predictable nucleotide substitutions and targeted deletion and insertions, greatly expanding the scope of genome editing [14,15,16,17,18,19,20]. The prerequisite for applying these technologies is delivery of the CRISPR/Cas reagents into plant cells. Usually, they are delivered into plant cells using *Agrobacterium*-mediated transformation, particle bombardment, protoplast transfection, or viral vectors [21,22,23].

Plant viruses are obligate intracellular pathogens that have been developed both as vectors for heterologous protein expression and as research tools for gene functional studies in plants. Virus-induced gene regulation systems in plants include the virus-mediated overexpression of heterologous proteins, virus-induced gene silencing, and host-induced gene silencing of pathogens or pests. These systems provide rapid, convenient, and high-throughput research tools for functional genomic studies [24,25]. Recently, many plant viral vectors have been successfully developed to deliver CRISPR/Cas reagents to both model and non-model plants, and virus-induced gene editing (VIGE) has been used to edit plant genomes. VIGE offers significant advantages over other technologies, including higher editing efficiency, accuracy, and operability [26]. In this review, we first describe CRISPR/Cas systems as genome editing tools. Next, we focus on VIGE and the types of viruses used to deliver CRISPR/Cas reagents into plant cells, comparison of different types of viral vectors for VIGE in plants, and the potential bottlenecks and future prospects of VIGE.

## 2. CRISPR/Cas: A Useful Genome Editing Tool

CRISPR/Cas systems confer adaptive immunity against invading elements in archaea and bacteria [1]. The defense process can be separated into three stages: CRISPR spacer acquisition, CRISPR expression, and CRISPR interference [12,27]. When spacers in the CRISPR RNA (crRNA) pair perfectly with an invasive nucleic acid, it initiates cleavage of the invading DNA by Cas proteins. Therefore, CRISPR/Cas systems provide dynamic and effective immunity against invading genetic elements. Sequence-specific RNA–DNA recognition and cleavage by Cas nucleases make CRISPR/Cas systems useful as programmable genome editing tools [28].

Type II CRISPR/Cas is the simplest and most well-studied system by far. It requires a single protein, Cas9, which is guided by paired trans-activating crRNA (tracrRNA) and crRNA molecules to introduce site-specific double-stranded breaks (DSBs) into a target DNA sequence during the interference stage. The type II CRISPR/Cas system from *Streptococcus pyogenes* has been developed as an RNA-programmable genome editing tool CRISPR/Cas9 [2]. A sgRNA engineered from a dual tracrRNA:crRNA molecule directs Cas9 to the target site. Then, Cas9 utilizes two distinct nuclease domains, HNH and RuvC-like, to cleave both strands of the target DNA, generating sequence-specific DSBs. This triggers two DNA repair systems, nonhomologous end-joining (NHEJ) and homology-directed repair (HDR) [28]. NHEJ, which is prevalent in most cells, is an error-prone pathway. It generates random indels at the break and thus frequently produces knockout mutants. In comparison, HDR is a less-frequent but high-fidelity pathway. If a homologous DNA template surrounds the break site, it may be repaired through HDR, generating precisely desired modifications such as insertions, replacements, and point mutations [3,29].

The current CRISPR/Cas9 system has been simplified and repurposed such that it has two components: the Cas9 nuclease and sgRNA. Target recognition by Cas9 requires both a 20-nucleotide sequence at the 5′ end of the sgRNA and a NGG protospacer-adjacent motif (PAM). By changing the nucleotides in the sgRNA, desired mutants can be obtained. Since the first application of CRISPR/Cas9 in plants [10,30,31], this technology has become a mainstream gene editing tool and has been applied to a variety of plant species for the functional annotation of genomes and genetic improvement of crops [4,15].

Genome editing technology is evolving rapidly. CRISPR/Cas12a, a type V CRISPR system, has also been adapted for genome editing, including in plants [32]. Moreover, recently developed base editors and prime editors allow for precise genome modifications. Collectively, these technologies have already had an impact on plant biological research and will play an important role in crop breeding [5,6,7,8].

The effective application of CRISPR/Cas technologies in plants requires delivery of the CRISPR/Cas cassette into plant cells (Figure 1). CRISPR/Cas reagents, including DNA, RNA, and ribonucleoprotein (RNP), are usually delivered into plant cells by *Agrobacterium*-mediated transformation, particle bombardment, or protoplast transfection [21,22,23]. However, these widely used delivery methods have limitations. For instance, protoplast transfection is normally used for transient expression, whereas particle bombardment requires expensive and specialized equipment, and it is restricted by a complex procedure and low efficiency. Similarly, the efficiency of *Agrobacterium*-mediated transformation is dependent on genotype, with few cultivars being amenable to transformation. Furthermore, none of these methods avoids complex tissue culture procedures. As an alternative to these methods, plant viruses have been harnessed to deliver CRISPR/Cas reagents into plant cells.

## 3. VIGE and Its Application in Plants

After nearly 10 years of development, VIGE systems utilizing numerous DNA and RNA viruses have been developed and applied to a variety of host plants, with excellent outcomes in genome editing (Table 1). VIGE has several advantages, including (a) multiple virus species or variants that may be combined with different host species or genotypes; (b) a high copy number of sgRNAs, which improves the editing efficiency; (c) rapid infection, leading to faster acquisition of the edited genotype; and (d) no integration of exogenous DNA, thus reducing off-target effects [33,34]. VIGE vectors can be classified into two categories according to their cargo capacity and the reagents that may be delivered (Figure 2). The first category includes VIGE vectors that express a sgRNA, infect plants that stably express Cas9, and yield gene-edited Cas9 transgenic seeds. The progeny can then be crossed with wild-type plants to remove the Cas9-encoding gene [35,36]. The second category includes VIGE vectors that deliver both Cas9 and the sgRNA, and spread systemically in plants. It requires tissue culture from infected leaves to regenerate gene-edited plants, which subsequently produce gene-edited seeds [37,38,39].

### 3.1. Viral Vectors Expressing sgRNAs

Due to a limitation in the carrying capacity of foreign genes, VIGE strategies are mainly focused on the delivery of sgRNAs to plants that constitutively overexpress Cas9. This not only ensures a systemic viral infection, it also increases the sgRNA concentration [40]. Currently, several plant viruses have been tested as vectors for the delivery of sgRNAs to create targeted knockouts, including tobacco rattle virus (TRV), potato virus X (PVX), cotton leaf crumple virus (CLCrV), barley stripe mosaic virus (BSMV), foxtail mosaic virus (FoMV), tobacco mosaic virus (TMV), pea early browning virus (PEBV), and beet necrotic yellow vein virus (BNYVV) (Table 1).

TRV (genus *Tobravirus*, family *Virgaviridae*) is a positive single-stranded RNA (+ssRNA) virus with a bipartite genome composed of RNA1 and RNA2 [60]. TRV has a wide host range, and its genome can be easily manipulated. Therefore, it has been widely used as a VIGE vector in functional genomic studies in plants [61]. Recently, a TRV RNA2 genome-derived vector was constructed and optimized for sgRNA delivery. After co-infiltrated the engineered TRV RNA2 with sgRNAs and RNA1 genome into *N. benthamiana* overexpressing Cas9, single or multiplex editing of targeted genes was achieved. Germinal transmission was detected. However, it was only in progeny seed from early flowers, indicating that the TRV infection and persistence in meristematic cells need to be optimized [40]. Further studies have demonstrated that TRV can successfully edit the *AtGL1* and *AtTT4* genes in *Arabidopsis* [34]. To improve the heritability of the TRV-based VIGE, Ellison et al. fused *Arabidopsis* FLOWERING LOCUS T (FT) mRNA to the 3′ end of sgRNA. It promotes the entry of sgRNAs to reproductive organs. Through expressing sgRNAs augmented with mobile RNA sequences, high editing efficiency (90–100%) was observed in infected tissues. More importantly, the efficiency of the inheritable genome edits (65–100%) was significantly increased [33]. Moreover, TRV-based sgRNA delivery systems function quickly and efficiently in terms of transcriptional activation and epigenomic editing [62].

PVX is the type member of the genus *Potexvirus* that infects 62 plant species, including important crops in the family *Solanaceae*. The PVX virion is a flexuous rod with a 6345-nucleotide (+) ssRNA genome, which is frequently manipulated and used as a plant RNA virus expression vector [41]. Recently, PVX was engineered to a vector expressing one or more sgRNAs in solanaceous plants. This PVX-based vector successfully delivered one or several sgRNAs into Cas9 transgenic *N. benthamiana*, achieving highly efficient multiplex editing in adult plant tissues. Furthermore, whole plants carrying indels at the target genes were regenerated from PVX infected tissues. This PVX VIGE vector allows efficient and multiplex genome editing and will be a useful tool for functional genomics and breeding in important crops of the family *Solanaceae* [41].

Cotton leaf crumple virus (CLCrV) is a two-component DNA virus composed of the CLCrV-A and -B genome. It was developed to deliver sgRNAs into Cas9 overexpression *Arabidopsis*. This CLCrV-mediated VIGE enabled targeted editing of endogenous genes in *Arabidopsis*. Furthermore, sgRNAs fused with FT mRNA at the 5′ end enabled effective and heritable gene editing, with an efficiency of 4.35–8.79% [47]. This allowed heritable gene editing avoiding tissue culture and stable transformation in *Arabidopsis*, suggesting broad application prospects in crops.

Genome editing in plants usually relies on conventional genetic transformation and regeneration procedures, which can be inefficient. However, virus-mediated sgRNA delivery systems show advantages in monocots such as maize and wheat. Barley stripe mosaic virus (BSMV) is a positive-sense RNA virus with three genome components (designated alpha, beta, and gamma) that infects many economically important monocot species. A BSMV-based sgRNA delivery system was developed that greatly simplifies CRISPR/Cas9-based gene editing in wheat and maize [45]. Due to continuous improvement and upgrades, the BSMV-based sgRNA delivery vector is capable of highly efficient, heritable genome editing in Cas9-transgenic wheat. The BSMV vector can carry multiple sgRNAs, and multiplex mutagenesis has been reported in the progeny. Furthermore, BSMV-infected Cas9-transgenic wheat pollen grains were crossed with wild-type wheat, leading to F1 progeny. After selfing the F1 mutants, Cas9-free wheat mutants were generated [43]. In one recent study, the BSMV-mediated sgRNA delivery system was used to edit *TaHRC*, improving Fusarium head blight resistance in wheat with no genotype limitation [44].

Foxtail mosaic virus (FoMV) is another useful viral vector in monocots [63]. FoMV successfully delivered a functional sgRNA into Cas9 transgenic maize. The sgRNA expressed from a duplicated promoter mediated successful edits in the maize *HKT1* gene. Moreover, the efficiency of editing could be enhanced in the presence of synergistic viruses and a viral silencing suppressor. However, because the edited plants described above were grown in a growth chamber and exhibited severe symptoms, the heritability of the mutations was not tested [48].

Overall, the combination of VIGE and transgenic Cas9 plants has resulted in high editing efficiency and simplified the process of generating gene-edited plants.

### 3.2. Viral Vectors That Can Express Both Cas9 and a sgRNA

Although the delivery of sgRNAs with viral vectors could induce high-frequency gene editing in plants that constitutively overexpress Cas9, delivery of the entire CRISPR/Cas reagents with viral vectors would be ideal to avoid the process for transgene entirely. Some viruses, with large cargo capacities and high gene stability, have been tested as vectors for the delivery of Cas9 and sgRNAs into plants to generate targeted knockouts, including PVX, barley yellow striate mosaic virus (BYSMV), and sonchus yellow net virus (SYNV) (Table 1).

The PVX virion is a flexuous rod. The filamentous, flexible structure of the PVX makes it unlikely that gene insert size is physically limited in the PVX vector [64]. Therefore, the entire CRISPR/Cas9 cassette can be inserted into the PVX vector. Following agroinoculation, targeted gene mutagenesis and base editing were successfully achieved in *N. benthamiana*. The genome editing efficiency (62%) in PVX-inoculated leaves was markedly higher than that in routine binary expression vector-agroinfiltrated leaves; the T-DNA integration rate was only 18%, which is a much lower acquisition rate than in the transgene-free genome-edited plants [37].

To avoid the possible integration of exogenous DNA into the plant genome during gene editing, mechanical inoculation with virions based on the PVX vector was applied to *N. benthamiana* leaves. Genome-edited regenerated shoots were obtained. Although the efficiency (~3%) was much lower than that with agroinoculation, this highlights the great potential of a PVX-based system for transgene-free gene editing in commercially important crops such as potato, tomato, eggplant, and pepper, which are natural hosts of PVX [37].

FoMV, another virus in the genus *Potexvirus*, has been used as an alternative expression vector for CRISPR/Cas reagents. First, Cas9 and sgRNA sequences were cloned into the FoMV vector respectively. Following the co-agroinfiltration of germinating *N. benthamiana* seeds, the plants developed evident systemic viral symptoms, and successful targeted mutagenesis of *PDS* was obtained. Moreover, co-delivery of the gene silencing suppressor p19 markedly enhanced the levels of Cas9 in young leaf tissues and led to efficient systemic gene editing of the target. However, no data showing heritable editing were obtained; this still needs to be further improved [49].

Mixed-infection with two or more viruses is frequently used to study virus–plant host/vector interactions. Recently, it has also been applied to construct VIGE vectors for genome editing. PVX and (tobacco etch virus, TEV) are two compatible viruses that can replicate and proliferate in the same host cells. The TEV was used to express Cas12a nuclease by replacing the *NIb* gene in the TEV genome. Another PVX virus vector expressing both sgRNAs and the TEV-deleted *NIb* gene was constructed. This dual virus-based vector system exhibited 20% indels in wild-type *N. benthamiana* [51]. This novel two compatible RNA virus-based system broadens the toolbox of VIGE.

Plant negative-stranded RNA (NSR) viruses have not been extensively used as expression vectors because of difficulty in engineering infectious cDNA clones [65,66]. However, compared to positive-strand RNA or DNA viruses, NSR viruses have large cargo capacities and high gene stability, making them suitable candidates for expressing large foreign sequences (e.g., CRISPR/Cas9 cassette). To date, two NSR viruses, BYSMV and SYNV, have been engineered to deliver the complete CRISPR/Cas9 cassette into plant cells for genome editing [38,39].

BYSMV belongs to the genus *Cytorhabdovirus*. It has been used as a model to develop the first recombinant cytorhabdovirus from cloned cDNAs. The BYSMV vector was also engineered and developed to simultaneously deliver Cas9 and sgRNAs to *N. benthamiana*. Various indels were obtained at the target site in the infected leaves, indicating that BYSMV-based vectors could mediate genome editing in *N. benthamiana*. However, it did not work in cereal plants [38].

SYNV, belonging to the genus *Nucleorhabdovirus*, family *Rhabdoviridae*, is the first plant NSR virus used to establish a reverse genetics system with cloned cDNAs [66]. SYNV has been used to simultaneously express Cas9 and sgRNAs in *N. benthamiana*. SYNV-mediated genome editing can generate single mutations, multiplex mutagenesis, and chromosome deletions with high efficiency. The viral vector remains stable during systemic infection or even after mechanical transmission, and it can be eliminated from edited plants during regeneration or after seed-setting. DNA-free and virus-free genome-edited plants have been regenerated from symptomatic upper leaf tissues. Although the restricted host range of SYNV limits its application to various species, it is a robust DNA-free method for generating heritable edits through leaf inoculation [39].

### 3.3. Geminivirus-Based Replicons for Genome Engineering

To increase their cargo capacity, some viral vectors have been deconstructed into non-infectious replicons (GVRs) by deleting non-replication-related genes, including movement protein- and coat protein-coding sequences; thus, they are not infectious on their own [67].

Geminiviruses, single-stranded circular DNA viruses of the family *Geminiviridae*, are widely distributed, transmitted by insects, and can infect a variety of plants worldwide [68]. Geminiviruses require only one replication-associated protein to achieve replication in a host cell; they also replicate efficiently and produce a high copy number of replicons [69]. Using the GVR strategy, only replication-associated blocks are kept. Geminivirus-based replicons can be used to deliver CRISPR/Cas9 cassette and supply adequate donor repair templates to facilitate gene targeting (GT).

The first Geminivirus vector was developed based on bean yellow dwarf virus (BeYDV). It was used to express CRISPR/Cas9 components and repair templates for genome editing in tobacco. The resulting targeted point mutations in the endogenous *ALS* gene enabled the regenerated tobacco plants to acquire herbicide resistance, and the efficiency was higher than that with conventional *Agrobacterium*-mediated transformation [52]. The BeYDV replicon was also used to deliver CRISPR/Cas reagents targeting *ALS1* and repair templates in potato. A targeted point mutation was obtained, and the regenerated transformed plantlets exhibited a reduced herbicide susceptibility phenotype [53]. Furthermore, the BeYDV vector was used to precisely integrate the strong 35S promoter upstream of an anthocyanin synthesis gene (*ANT1*) in tomato. Heritable modification was obtained at frequencies 10-fold higher than with traditional DNA delivery methods (i.e., *Agrobacterium*) [54].

Geminivirus replicon-mediated genome editing has also been successfully achieved in monocots. A deconstructed version of wheat dwarf virus (WDV) was developed for genome editing in cereal crops. In one study, WDV replicons carrying both CRISPR/Cas cassette and a donor template achieved GT at an endogenous wheat locus at frequencies 12-fold higher than that with non-viral delivery methods. Targeted integration by HDR was achieved in all three of the homoeoalleles (A, B, and D). Additionally, multiplexed GT within the same cell was achieved at frequencies of 1% [56]. Similarly, by combining CRISPR/Cas9 and geminiviral vectors, precise and efficient DNA knock-in mutants have been generated in rice [57].

Geminivirus-based replicons enable highly efficient genome engineering. Compared to DNA virus replicons, RNA virus vectors have the advantage of completely avoiding integration into plant genomes and producing exogenous DNA-free plants, thus avoiding raising additional safety concerns.

### 3.4. Comparison of Different Types of Viral Vectors for VIGE in Plants

Due to a limitation in the carrying capacity of foreign genes, VIGE vectors are mainly focused on the delivery of sgRNAs to plants, which need to be in conjunction with Cas9-overexpressing host plants to induce genome editing. Targeted genome editing occurs in individual plant cells, which can be used for regeneration. Then genome-edited plantlets are generated through screening. Alternatively, sgRNAs spread systemically in the plants, generating inheritable mutagenesis. Although these gene-edited seeds harbor Cas9 overexpressing, it can be segregated out through backcrossing [43]. Currently, viral vectors expressing sgRNAs are widely used in both dicots and monocots (Table 1).

Viral vectors expressing both Cas9 and sgRNAs are ideal to avoid the process for transgene entirely. However, it requires viruses to have large genomes and flexible virions, and only a few are available (Table 1). These viruses (e.g., BYSMV, SYNV) have restricted host ranges [38,39], limiting its application to various species. With further exploration, those types may be widely used because of their large carrying capacity and convenient operation.

Geminiviruses are widely distributed and infect a variety of plants. To express both Cas9 and sgRNAs, non-replication-related genes are deleted. Geminivirus-based replicons can deliver CRISPR/Cas9 cassette and supply adequate repair templates to improve gene editing efficiency and facilitate gene targeting [54]. However, this strategy is completely dependent on *Agrobacterium* delivery and tissue culture.

## 4. Challenges and Perspectives for VIGE in Plants

As described above, VIGE systems offer several advantages, including an improved editing efficiency by increasing sgRNA expression, time savings by avoiding lengthy and laborious tissue culture procedures in favor of invading and editing meristem cells directly, and the rapid acquisition of gene editing-induced phenotypes through viral infection. Additionally, VIGE largely avoids off-target effects, especially RNA virus-mediated VIGE systems, since they do not integrate into the host genome.

However, bottlenecks and defects in current VIGE systems exist. The biggest problem is that the cargo capacities of plant DNA or RNA viruses are limited [33,43]. The greater the length of the foreign gene, the less stable the viral genome is; Cas9 cannot be co-encoded with the sgRNA by such viruses. Therefore, stable transgenic Cas9-overexpressing lines are required and they must be infected with RNA viral vectors expressing a modified sgRNA for genome editing [33,43,47,50,70]. Moreover, few RNA viruses with excellent genome stability and a high delivery capacity can be engineered to deliver the entire CRISPR/Cas9 cassette to achieve genome editing in plants [37,38,39]. Recently, smaller RNA-guided nucleases such as Cas12e (986 amino acids), Cas12j (700–800 amino acids), and Cas12f (400–600 amino acids) have been demonstrated to introduce site-specific DSBs in target DNAs and characterized as useful genome editing tools for eukaryotic cells [71,72,73]. Although their efficiency should be further optimized, these miniature nucleases may help to overcome viral genome packaging constraints. A VIGE system employing these newly identified compact Cas proteins should be explored for plant genome editing.

Another defect is that intact viruses cannot enter meristem cells or reproductive tissues, making it difficult to achieve heritable gene editing in plants. To solve this problem, researchers have cleverly added endogenous mobile RNA sequences (e.g., FT or tRNAs) to the 3′ end of sgRNAs [33]. These mobile elements have greatly increased the movement of sgRNAs into the shoot apical meristem, resulting in heritable editing without tissue culture [43,44]. However, this strategy requires Cas9-overexpressing lines, and it improves the mobility of the sgRNA but not Cas9 into meristem cells. Therefore, the development of viral vectors with a large genome cargo size and the ability to infect meristem or germline cells is still needed.

CRISPRed DNA-free plants are not different from those generated by natural or random mutagenesis. Therefore, they may accelerate the process of crop breeding. In many countries such as the United States, Japan, and Australia, gene-edited DNA-free crops are exempt from restrictive policies applied to GMO legislation [74]. Recently, China’s Ministry of Agriculture and Rural Affairs has issued a guideline for the regulatory approval of gene-edited crops. According to the guideline, gene-edited DNA-free crops may require much less complicated safety evaluations compared to GMOs and may contribute to sustainable agriculture and grain security in China. Recently, a negative-strand RNA virus delivering the entire CRISPR/Cas9 cassette was used to generate DNA-free genome-edited tobacco [39]. This viral delivery system seems to be the most convenient method to obtain CRISPRed DNA-free plants; however, the apparent limitation is the viral host range. Given the reverse genetic tools available for an increasing number of similar viruses, this strategy may be applicable to other rhabdoviruses that infect diverse crop species.

## 5. Conclusions

CRISPR/Cas-based genome editing technologies have revolutionized plant science and enabled the creation of germplasms with beneficial traits. Currently, numerous plant viruses have been engineered to deliver CRISPR/Cas reagents into plant cells, with excellent outcomes in genome editing in a variety of host plants. Although there are some bottlenecks and defects in current VIGE systems, we expect that an ideal VIGE system with high efficiency, which is DNA-free, and which shows strong heritability will be developed with further exploration. We also believe that the use of viruses to deliver CRISPR/Cas reagents will hold enormous promise for creating elite crop varieties that will promote future global food security.

## Figures and Tables

**Figure 1 ijms-23-10202-f001:**
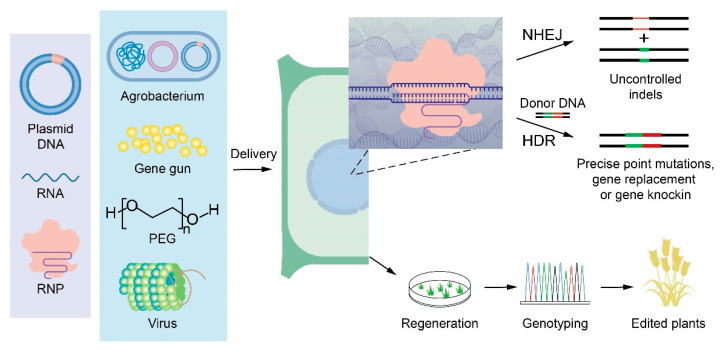
Schematic illustration of the major steps in plant genome editing. DNA, RNA-encoding CRISPR/Cas reagents, or RNPs (composed of Cas9 and an in vitro-transcribed sgRNA) can be delivered into plant cells using *Agrobacterium* cells, a gene gun, polyethyleneglycol, or viruses. In the nucleus, the CRISPR/Cas reagent creates site-specific DSBs, which may be repaired through the NHEJ or HDR pathways. NHEJ generates uncontrolled, but predictable indels. In the presence of a donor template, breaks may be repaired through HDR, generating precise modifications. Gene-edited plants are identified from among the regenerated plants through genotyping.

**Figure 2 ijms-23-10202-f002:**
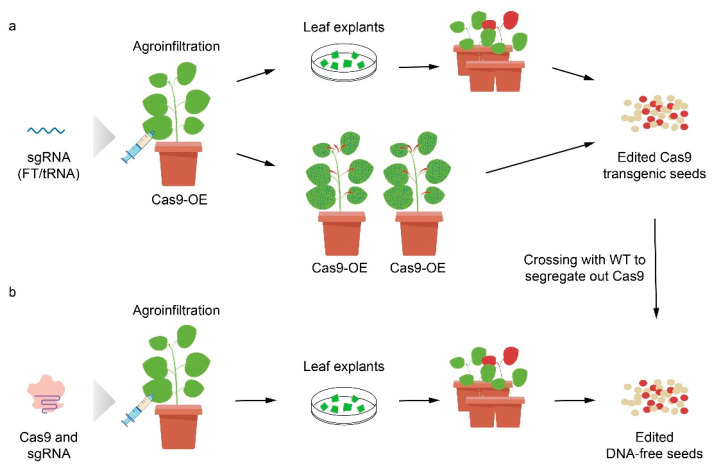
Overview of the methods available to create heritable edits in plants. (**a**) VIGE through sgRNA expression. *Agrobacterium* cells carrying viral vectors expressing sgRNAs are agroinfiltrated into the leaves of Cas9-overexpressing plants. The CRISPR/Cas9 complex induces targeted genome editing in individual plant cells, which can be used for regeneration. Alternatively, genome editing occurs as the virus moves systemically throughout the plant. Gene-edited Cas9 transgenic seeds are then obtained. After crossing with wild type (WT), Cas9 can be segregated out, and gene-edited DNA-free seeds are produced. (**b**) VIGE through Cas9 and sgRNA expression. *Agrobacterium* cells carrying viral vectors expressing Cas9 and sgRNAs are agroinfiltrated into the leaves of wild-type plants. The CRISPR/Cas9 complex induces targeted genome editing in individual plant cells, which can be used for regeneration. Gene-edited DNA-free seeds are produced by the regenerated plants.

**Table 1 ijms-23-10202-t001:** Virus delivering CRISPR/Cas9 reagents or sgRNAs to plant cells for genome editing.

Virus	Receptor Plants	Virus Insert Cargo	Target(s)	Tissue Culture	Mutations Heritable	Refs.
Tobacco rattle virus (TRV)	Cas9-expressing *N. benthamiana*	sgRNAs (+/−FT)	*NbPDS*, *NbAG*	No	Yes	[33]
TRV	Cas9-expressing *Arabidopsis*	sgRNAs	*AtGL1*, *AtTT4*	No	No	[34]
TRV	Cas9-expressing *N. benthamiana*	sgRNAs	*NbPDS3*, *NbPCNA*	No	Yes	[35]
TRV	Cas9-expressing *Arabidopsis*	sgRNAs (+/−FT or tRNA)	*AtFWA*	No	Yes	[36]
TRV	Cas9-expressing *N. benthamiana*	sgRNAs	*NbPDS3*	No	Yes	[40]
Potato virus X (PVX)	Cas9-expressing *N. benthamiana*	sgRNAs (+/−FT or tRNA)	*NbXT2B*, *NbPDS*, *NbFT*	Yes	Yes	[41]
PVX	*N. benthamiana*	Cas9 and sgRNAs	*NbTOM1*	Yes	Yes	[37]
Barley yellow striate mosaic virus (BYSMV)	GFP-expressing *N. benthamiana*	Cas9 and sgRNAs	*GFP*	No	No	[38]
Sonchus yellow net rhabdovirus (SYNV)	*N. benthamiana* (WT or GFP expressing)	Cas9 and sgRNAs	*GFP*, *NbPDS*, *NbRDR6*, *NbSGS3*	Yes	Yes	[39]
Pea early browning virus (PEBV)	Cas9-expressing *N. benthamiana*	sgRNAs	*NbPDS*	No	No	[34]
Apple latent pherical virus (ALSV)	Cas9-expressing *N. benthamiana* soybean	sgRNAs	*NbPDS*, *EPSPS*, *GmGW2*	No	No	[36]
Barley stripe mosaic virus (BSMV)	Cas9-expressing wheat	sgRNAs (+/−FT or tRNA)	*TaGW2*, *TaUPL3*, *TaGW7*, *TaQ*	No	Yes	[42]
BSMV	*N. benthamiana*; Cas9-expressing wheat	sgRNAs (+/−FT or tRNA)	*TaPDS*, *TaGW2*, *TaGASR7*	No	Yes	[43]
BSMV	*N. benthamiana*; Cas9-expressing wheat	sgRNAs (+/−FT or tRNA)	*TaHRC*	No	Yes	[44]
BSMV	*N. benthamiana* (WT or GFP expressing); Cas9-expressing wheat/maize	sgRNAs	*NbPDS*, *GFP* GASR7, TMS5	Yes	Yes No	[45]
Beet necrotic yellow vein virus (BNYVV)	Cas9-expressing *N. benthamiana*	sgRNAs	*NbPDS3*	No	No	[46]
Cotton leaf crumple virus (CLCrV)	Cas9-expressing *Arabidopsis*	sgRNAs (+/−FT)	*AtBRI1*, *AtGL2*	No	Yes	[47]
Foxtail mosaic virus (FoMV)	Cas9-expressing *N. benthamiana S.viridis*, Maize	sgRNAs	*NbPDS* *SvCA2*, *ZmHKT1*	No	No	[48]
FoMV	*N.benthamiana*	Cas9, sgRNAs (+/−P19)	*NbPDS*	No	No	[49]
Cabbage Leaf Curl virus (CaLCuV)	Cas9-expressing *N. benthamiana*	sgRNAs	*NbIspH*, *NbPDS*	Yes	No	[50]
Tobacco etch virus (TEV) and PVX	*N. benthamiana*	Cas12a and sgRNAs, respectively	*NbXT1*, *NbFT*	No	No	[51]
Bean yellow dwarf virus (BeYDV)	Tobacco	Cas9 and sgRNAs	*AtADH1*	Yes	Yes	[52]
BeYDV	Potato	Cas9 and sgRNAs	*StALS1*	Yes	Yes	[53]
BeYDV	Tomato	Cas9 and sgRNAs	*SlANT1*	Yes	Yes	[54]
BeYDV	Tomato	Cas9 and sgRNAs	*SlCRTISO*, *SlPSY1*	Yes	Yes	[55]
Wheat dwarfing virus (WDV)	Wheat	Cas9 and sgRNAs	*GFP*, *BFP*	Yes	Yes	[56]
WDV	Rice	Cas9 and sgRNAs	*GFP*	Yes	Yes	[57]
Tobacco mosaic virus (TMV)	GFP-expressing *N. benthamiana*	Cas9 and sgRNAs	*GFP*, *NbAGO1*	No	No	[58]
TMV	GFP-expressing *N. benthamiana*	Cas9 and sgRNAs	*GFP*	No	No	[59]

## Data Availability

The datasets generated during and/or analyzed during the current study are available from the corresponding author on reasonable request.

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
