# Peer review of "Virus-Induced Gene Editing and Its Applications in Plants"

_ijms, 2022, doi:10.3390/ijms231810202_

Round 1
Reviewer 1 Report
Review
Title: Virus-induced gene editing and its applications in plants
Type: Review paper
Journal: International Journal of Molecular Sciences – MDPI
Overview
This paper describes CRISPR/Cas systems as genome editing tools, focusing on how plant viruses can be engineered to deliver CRISPR/Cas reagents into plant cells, the advantages, and limitations of using these plant viral vectors, and the possibilities of VIGE. The manuscript is well written and brings an interesting/hot subject, the possibility of being a contribution to the scientific community. However, some issues must be addressed:
General observations:
• The absence of line numbers makes the review comments and suggestions very difficult.
• There are 4 and 10 references from 2021 and 2022, respectively, out of 81, which makes questionable its level of updates.
Few formatting issues:
• “Geminivirus-based replicons enable highly efficient genome engineering”. It is in different/smaller font size than the rest of the manuscript.
• There are “years” in references that are in bold, and some are not.
Related to the subject:
• “which allows the precise manipulation of plant genes”… Instead, of “gene” use “genome”.
• “however, this is limited by the rigid plant cell wall”… This is not the only limitation and depending on the methods used, all tissue culture challenges apply.
• “Here, we briefly describe CRISPR/Cas-based genome editing.” Currently, there are several good and recent reviews on this subject [1–4].
1. Impens, L.; Jacobs, T.B.; Nelissen, H.; Inzé, D.; Pauwels, L. Mini-Review: Transgenerational CRISPR/Cas9 Gene Editing in Plants. Front. Genome Ed. 2022, 4, 1–6, doi:10.3389/fgeed.2022.825042.
2. Son, S.; Park, S.R. Challenges Facing CRISPR/Cas9-Based Genome Editing in Plants. Front. Plant Sci. 2022, 13, doi:10.3389/fpls.2022.902413.
3. Li, Y.; Li, W.; Li, J. The CRISPR/Cas9 Revolution Continues: From Base Editing to Prime Editing in Plant Science. J. Genet. Genomics 2021, 48, 661–670, doi:10.1016/j.jgg.2021.05.001.
4. Hassan, M.M.; Zhang, Y.; Yuan, G.; De, K.; Chen, J.-G.; Muchero, W.; Tuskan, G.A.; Qi, Y.; Yang, X. Construct Design for CRISPR/Cas-Based Genome Editing in Plants. Trends Plant Sci. 2021, 26, 1133–1152, doi:10.1016/j.tplants.2021.06.015.
• “We then introduce VIGE systems”. Currently, there are several reviews on this subject that also includes CRISPR/Cas-based genome editing. Additionally, “introducing” a subject will be first presented and this is not the case here. [5–7]
5. Gentzel, I.N.; Ohlson, E.W.; Redinbaugh, M.G.; Wang, G.-L. VIGE: Virus-Induced Genome Editing for Improving Abiotic and Biotic Stress Traits in Plants. Stress Biol. 2022, 2, 2, doi:10.1007/s44154-021-00026-x.
6. Rössner, C.; Lotz, D.; Becker, A. VIGS Goes Viral: How VIGS Transforms Our Understanding of Plant Science. Annu. Rev. Plant Biol. 2022, 73, 703–728, doi:10.1146/annurev-arplant-102820-020542.
7. Oh, Y.; Kim, H.; Kim, S.-G. Virus-Induced Plant Genome Editing. Curr. Opin. Plant Biol. 2021, 60, 101992, doi:10.1016/j.pbi.2020.101992.
The introduction is concise and clear, highlighting the main points. However, all other topics are very superficial in addressing the subjects and missing figures to better illustrate the importance and complexity of the theme. We must consider that several other strong and complete reviews on the subject have been recently published in important journals, such as the Annual Review of Plant Biology and Trends in Plant Science. The point here is that nowadays there is a flood of review papers, which requires the selection of the best quality ones. Additionally, this manuscript is very short, superficial, and does not present enough, and significant new information to be published in such an important journal as IJMS, but, with some improvement, this manuscript could suitable for publication in another and less impactful journal.
Author Response
Overview: This paper describes CRISPR/Cas systems as genome editing tools, focusing on how plant viruses can be engineered to deliver CRISPR/Cas reagents into plant cells, the advantages, and limitations of using these plant viral vectors, and the possibilities of VIGE. The manuscript is well written and brings an interesting/hot subject, the possibility of being a contribution to the scientific community. However, some issues must be addressed:
General observations:
Point 1: The absence of line numbers makes the review comments and suggestions very difficult.
Response 1: We are sorry for that, and we have added line numbers in the revised manuscript.
Point 2: There are 4 and 10 references from 2021 and 2022, respectively, out of 81, which makes questionable its level of updates.
Response 2: We agree with the reviewer. We have revised the manuscript, and updated the references (e.g. added 4 references from 2021, 5 references from 2022) in the revised manuscript.
Few formatting issues:
Point 3: “Geminivirus-based replicons enable highly efficient genome engineering”. It is in different/smaller font size than the rest of the manuscript.
Response 3: Thanks for pointing out that. We have corrected it in the revised manuscript.
Point 4: There are “years” in references that are in bold, and some are not.
Response 4: We have corrected them in the revised manuscript.
Related to the subject:
Point 5: “which allows the precise manipulation of plant genes”… Instead, of “gene” use “genome”.
Response 5: We have corrected it in the revised manuscript.
Point 6: “however, this is limited by the rigid plant cell wall”… This is not the only limitation and depending on the methods used, all tissue culture challenges apply.
Response 6: Thanks for pointing out that. We revised the sentence to “however, this is limited by the tissue culture challenges”, that would be more rigorous.
Point 7: “Here, we briefly describe CRISPR/Cas-based genome editing.” Currently, there are several good and recent reviews on this subject [1–4].
- Impens, L.; Jacobs, T.B.; Nelissen, H.; Inzé, D.; Pauwels, L. Mini-Review: Transgenerational CRISPR/Cas9 Gene Editing in Plants. Front. Genome Ed.2022, 4, 1–6, doi:10.3389/fgeed.2022.825042.
- Son, S.; Park, S.R. Challenges Facing CRISPR/Cas9-Based Genome Editing in Plants. Front. Plant Sci.2022, 13, doi:10.3389/fpls.2022.902413.
- Li, Y.; Li, W.; Li, J. The CRISPR/Cas9 Revolution Continues: From Base Editing to Prime Editing in Plant Science. J. Genet. Genomics2021, 48, 661–670, doi:10.1016/j.jgg.2021.05.001.
- Hassan, M.M.; Zhang, Y.; Yuan, G.; De, K.; Chen, J.-G.; Muchero, W.; Tuskan, G.A.; Qi, Y.; Yang, X. Construct Design for CRISPR/Cas-Based Genome Editing in Plants. Trends Plant Sci.2021, 26, 1133–1152, doi:10.1016/j.tplants.2021.06.015.
“We then introduce VIGE systems”. Currently, there are several reviews on this subject that also includes CRISPR/Cas-based genome editing. Additionally, “introducing” a subject will be first presented and this is not the case here. [5–7]
- Gentzel, I.N.; Ohlson, E.W.; Redinbaugh, M.G.; Wang, G.-L. VIGE: Virus-Induced Genome Editing for Improving Abiotic and Biotic Stress Traits in Plants. Stress Biol.2022, 2, 2, doi:10.1007/s44154-021-00026-x.
- Rössner, C.; Lotz, D.; Becker, A. VIGS Goes Viral: How VIGS Transforms Our Understanding of Plant Science. Annu. Rev. Plant Biol.2022, 73, 703–728, doi:10.1146/annurev-arplant-102820-020542.
- Oh, Y.; Kim, H.; Kim, S.-G. Virus-Induced Plant Genome Editing. Curr. Opin. Plant Biol.2021, 60, 101992, doi:10.1016/j.pbi.2020.101992.
Response 7: The reviewer is right. There are many reviews about CRISPR/Cas-based genome editing and VIGE, and we have cited reviews listed by the reviewer in the revised manuscript. Authors with different knowledge and professional backgrounds give readers different enlightenment. Here, we mainly focus on VIGE systems and the types of viruses used currently for CRISPR/Cas9 cassette delivery and genome editing. We summarize the advantages and disadvantages of utilizing these plant viral vectors, and the application of VIGE to a variety of host plants. We also discuss the potential bottlenecks and future prospects for VIGE in plants. Additionally, we corrected the sentence to “We then focus on VIGE systems” in the revised manuscript.
Point 8: The introduction is concise and clear, highlighting the main points. However, all other topics are very superficial in addressing the subjects and missing figures to better illustrate the importance and complexity of the theme. We must consider that several other strong and complete reviews on the subject have been recently published in important journals, such as the Annual Review of Plant Biology and Trends in Plant Science. The point here is that nowadays there is a flood of review papers, which requires the selection of the best quality ones. Additionally, this manuscript is very short, superficial, and does not present enough, and significant new information to be published in such an important journal as IJMS, but, with some improvement, this manuscript could suitable for publication in another and less impactful journal.
Response 8: Thanks for the suggestion. In the revised manuscript, we took the suggestion, enriched Part 3, highlighted significant works to better illustrate the importance and complexity of VIGE, added the comparison between the types of viral vectors and their advantages/disadvantages of use (Part 3.4), gave some perspective of this technology for crop breeding (Part 4), and cited review papers (Rössner et al. 2022, Annu. Rev. Plant Biol. 73, 703–728; Hassan et al. 2021, Trends Plant Sci. 26, 1133–1152) listed by the reviewer.

Reviewer 2 Report
The review, which describes the possibilities of engineering plants in order to increase the quality of products but the productivity itself through the use of genome editing techniques seems to be quite clear. I would suggest you to enrich the comparison between the types of viral vectors and their advantages / disadvantages of use so that the discourse is clear to those who approach this issue for the first time. Good use of English language.
Author Response
Point 1: The review, which describes the possibilities of engineering plants in order to increase the quality of products but the productivity itself through the use of genome editing techniques seems to be quite clear. I would suggest you to enrich the comparison between the types of viral vectors and their advantages / disadvantages of use so that the discourse is clear to those who approach this issue for the first time. Good use of English language.
Response 1: Thanks for the suggestion, we have added those as suggested (Part 3.4 Comparison of different types of viral vectors for VIGE in plants) in the revised manuscript. It will be helpful to those who approach this issue for the first time.

Reviewer 3 Report
- There are already many reviews (*) on VIGE around: what is the added value of this review? Please explain
*) Examples:
Gong et al., 2021 (https://doi.org/10.3389/fgeed.2021.8172790),
Wang et al., 2020 (https://doi.org/10.3390/v12111338),
Oh et al., 2021 (https://doi.org/10.1016/j.pbi.2020.101992),
Gentzel et al., 2022 (https://doi.org/10.1007/s44154-021-00026-x),
Gong et al., 2021 (https://doi.org/10.3389/fgeed.2021.817279)
- New data and insights make up most of the conclusion section hence these are not truly conclusions drawn from the previous sections. Additional data and true conclusions should be separated from each other and divided into different sections
- Here and there some additional data that are not relevant, such as transmission of the BYSMV by planthoppers, the different types of natural CRISPR/Cas systems in bacteria , etc., which can be left out without affecting the overall gist and message of the article. Please delete these details.
- It is not explained that in many cases, it is not a true virus which is delivered to the plant cells but actually a cDNA encoding viral RNA, for example. Please explain this.
- DNA-free methods may be popular in China because of their exemption from GMO legislation. Please clarify this.
- Figure 2: HDR may also be used for precise point mutations, please adjust
- Many of the tabled examples have a host expressing Cas9. This would be regarded transgenic and would therefore go against the idea of transgenic
Author Response
Point 1: There are already many reviews (*) on VIGE around: what is the added value of this review? Please explain
*) Examples: Gong et al., 2021 (https://doi.org/10.3389/fgeed.2021.8172790), Wang et al., 2020 (https://doi.org/10.3390/v12111338),
Oh et al., 2021 (https://doi.org/10.1016/j.pbi.2020.101992), Gentzel et al., 2022 (https://doi.org/10.1007/s44154-021-00026-x),
Gong et al., 2021 (https://doi.org/10.3389/fgeed.2021.817279)
Response 1: The reviewer is right. There are currently many reviews on VIGE, and we have cited reviews listed by the reviewer in the revised manuscript. Authors with different knowledge and professional backgrounds give readers different enlightenment. Here, we mainly focus on VIGE systems and the types of viruses used currently for CRISPR/Cas9 cassette delivery and genome editing. We also summarize the advantages and disadvantages of utilizing these plant viral vectors, and the application of VIGE to a variety of host plants. Finally, we discuss the potential bottlenecks and future prospects for VIGE in plants.
Point 2: New data and insights make up most of the conclusion section hence these are not truly conclusions drawn from the previous sections. Additional data and true conclusions should be separated from each other and divided into different sections
Response 2: Thanks for pointing out that. We have separated the additional data (Part 4 Challenges and perspectives for VIGE in plants) and conclusions (Part 5), and rewrite the conclusions drawn from the previous sections in the revised manuscript.
Point 3: Here and there some additional data that are not relevant, such as transmission of the BYSMV by planthoppers, the different types of natural CRISPR/Cas systems in bacteria , etc., which can be left out without affecting the overall gist and message of the article. Please delete these details.
Response 3: We have checked the manuscript, and deleted the details that are not relevant as suggested in the revised manuscript.
Point 4: It is not explained that in many cases, it is not a true virus which is delivered to the plant cells but actually a cDNA encoding viral RNA, for example. Please explain this.
Response 4: The reviewer is right. Plant virus is an obligate parasite that can only exist in living cells. If the virus is used to express foreign genes, it can only be engineered to construct infectious cDNA clones. VIGE vectors are inoculated into plant cell. Then, inoculated leaves are used as viral sources for secondary inoculation. Those have been added in the revised manuscript.
Point 5: DNA-free methods may be popular in China because of their exemption from GMO legislation. Please clarify this.
Response 5: Gene-edited DNA-free plants are not different from those generated by natural or random mutagenesis, thus they may accelerate the process of crop breeding. Recently, China’s Ministry of Agriculture and Rural Affairs has issued a guideline for the regulatory approval of gene-edited crops. According to this guideline, gene-edited DNA-free crops may require much less complicated food and environmental safety evaluations compared to GMOs, and may contribute to sustainable agriculture and grain security. Therefore, gene-edited DNA-free methods may be popular in China. Those have been added in the revised manuscript.
Point 6: Figure 2: HDR may also be used for precise point mutations, please adjust
Response 6: We have adjusted the results of HDR in the revised Figure.
Point 7: Many of the tabled examples have a host expressing Cas9. This would be regarded transgenic and would therefore go against the idea of transgenic
Response 7: The reviewer is right. Due to viral genome packaging constraints, VIGE strategies are mainly focused on the delivery of sgRNAs to hosts expressing Cas9. Gene-edited Cas9 transgenic seeds are obtained. The progeny can then be crossed with wild-type plants to remove the Cas9-encoding gene. Finally, gene-edited DNA-free seeds are produced. We have adjusted Figure 2 in the revised manuscript.

Round 2
Reviewer 1 Report
Overview: This paper describes CRISPR/Cas systems as genome editing tools, focusing on how plant viruses can be engineered to deliver CRISPR/Cas reagents into plant cells, the advantages, and limitations of using these plant viral vectors, and the possibilities of VIGE. The manuscript is well written and brings an interesting/hot subject, the possibility of being a contribution to the scientific community. However, some issues must be addressed:
General observations:
Point 1: The absence of line numbers makes the review comments and suggestions very difficult.
Response 1: We are sorry for that, and we have added line numbers in the revised manuscript.
Reviewer response to point 2: Thank you!
Point 2: There are 4 and 10 references from 2021 and 2022, respectively, out of 81, which makes questionable its level of updates.
Response 2: We agree with the reviewer. We have revised the manuscript, and updated the references (e.g. added 4 references from 2021, 5 references from 2022) in the revised manuscript.
Reviewer response to point 2: When it was requested to add more recent refences to make this manuscript appealing and updated, it was meant to add more research papers related to the subject. Instead, 9 out of 12 references that authors added, after revisions, are reviews. These 9 reviews are listed bellow:
1. Hassan, M. M.; Zhang Y.; Yuan G.; De K.; Chen J. G.; Muchero W.; Tuskan G. A.; Qi Y.; Yang X. Construct design for CRISPR/Cas-based 405 genome editing in plants. Trends Plant Sci. 2021, 26(11), 1133-1152.
9. Son, S.; Park, S. R. Challenges facing CRISPR/Cas9-based genome editing in plants. Front Plant Sci. 2022, 13, 902413
12. Impens, L.; Jacobs T. B.; Nelissen H.; Inzé D.; Pauwels L. Mini-review: transgenerational CRISPR/Cas9 gene editing in plants. Front. 423 Genome Ed. 2022, 4, 825042.
13. Hsieh, Feng. V.; Yang, Y. Efficient expression of multiple guide RNAs for CRISPR/Cas genome editing. aBIOTECH 2020, 1, 123-134.
18. Li, J.; Yu X.; Zhang C.; Li N.; Zhao J. The application of CRISPR/Cas technologies to brassica crops: current progress and future perspectives. 433 aBIOTECH 2022, 3(2), 146-161.
22. Gong, Z.; Cheng M.; Botella J. R. Non-GM genome editing approaches in crops. Front. Genome Ed. 2021, 3, 817279.
24. Rössner, C.; Lotz D.; Becker A. VIGS goes viral: how VIGS transforms our understanding of plant science. Annu. Rev. Plant Biol. 2022, 73, 444 703-728.
77. Oh, Y.; Kim, H.; Kim, S. G. Virus-induced plant genome editing. Curr. Opin. Plant Biol. 2021, 60, 101992
82. He, Y.; Mudgett M.; Zhao, Y. Advances in gene editing without residual transgenes in plants. Plant Physiol. 2022, 188(4), 1757-1768
Apart from the above listed, this manuscript has so many more review papers in the references. The authors, themselves, show the flood of reviews on the subject by citing at least 22 review papers out of the former 81 in references in the first version of this manuscript. Please find these 22 below:
3. Gao, C. Genome engineering for crop improvement and future agriculture. Cell 2021, 184(6), 1621-1635.
4. Hua, K.; Zhang, J.; Botella, J. R.; Ma, C.; Kong, F.; Liu, B.; Zhu, J. K. Perspectives on the application of genome-editing technologies in 410 crop breeding. Mol. Plant 2019, 12(8), 1047-1059
5. Liu, G.; Lin, Q.; Jin, S.; Gao, C. The CRISPR-Cas toolbox and gene editing technologies. Mol. Cell 2022, 82(2), 333-347
7. Li, J.; Li, H.; Chen, J.; Yan, L.; Xia, L. Toward precision genome editing in crop plants. Mol. Plant 2020, 13(6), 811-813
8. Schindele, A.; Dorn, A.; Puchta, H. CRISPR/Cas brings plant biology and breeding into the fast lane. Curr. Opin. Biotechnol. 2020, 61, 7-
10. Shan, Q.; Wang, Y.; Li, J.; Zhang, Y.; Chen, K.; Liang, Z.; Zhang, K.; Liu, J.; Xi, J. J.; Qiu, J. L.; Gao. C. Targeted genome modification of 419 crop plants using a CRISPR-Cas system. Nat. Biotechnol. 2013, 31(8), 686-688
15. Pickar-Oliver, A.; Gersbach, C. A. The next generation of CRISPR-Cas technologies and applications. Nat Rev Mol. Cell Biol. 2019, 20(8), 428 490-50
16. Mao, Y.; Botella, J. R.; Liu Y.; Zhu, J. K. Gene editing in plants: progress and challenges. Natl. Sci. Rev. 2019, 6(3), 421-43
17. Li, Y.; Li, W.; Li, J. The CRISPR/Cas9 revolution continues: From base editing to prime editing in plant science. J. Genet. Genomics. 2021, 431 48(8), 661-670
23. Zhao, L.; Feng, C.; Wu, K.; Chen, W.; Chen, Y.; Hao, X.; Wu, Y. Advances and prospects in biogenic substances against plant virus: A 442 review. Pestic. Biochem. Physiol. 2017, 135, 15-26.
Salazar-González, J. A.; Bañuelos-Hernández, B.; Rosales-Mendoza, S. Current status of viral expression systems in plants and perspectives 446 for oral vaccines development. Plant Mol. Biol. 2015, 87(3), 203-217.
Abrahamian, P.; Hammond, R. W.; Hammond, J. Plant virus-derived vectors: applications in agricultural and medical biotechnology. Annu. 448 Rev. Virol. 2020, 7(1), 513-535
Wang, M.; Gao, S.; Zeng, W.; Yang, Y.; Ma, J.; Wang, Y. Plant virology delivers diverse toolsets for biotechnology. Viruses 2020, 12(11), 450 1338
28. Makarova, K. S.; Wolf, Y. I.; Iranzo, J.; Shmakov, S. A.; Alkhnbashi, O. S.; Brouns, S. J. J.; Charpentier, E.; Cheng, D.; Haft, D, H.; Horvath, 452 P.; et al. Evolutionary classification of CRISPR-Cas systems: a burst of class 2 and derived variants. Nat. Rev. Microbiol. 2020, 18(2), 67- 453 83
29. Chen, K.; Wang, Y.; Zhang, R.; Zhang, H.; Gao, C. CRISPR/Cas genome editing and precision plant breeding in agriculture. Annu. Rev. 455 Plant Biol. 2019, 70, 667-69
30. Rozov, S. M.; Permyakova N. V.; Deineko, E. V. The problem of the low rates of CRISPR/Cas9-mediated knock-ins in plants: approaches 457 and solutions. Int. J. Mol. Sci. 2019, 20(13), 337
31. Liu, Q.; Yang, F.; Zhang, J.; Liu, H.; Rahman, S.; Islam, S.; Ma, W.; She, M. Application of CRISPR/Cas9 in crop quality improvement. 459 Int. J. Mol. Sci. 2021, 22(8), 4206
34. Li, J.; Li, Y.; Ma, L. Recent advances in CRISPR/Cas9 and applications for wheat functional genomics and breeding. aBIOTECH 2021, 2(4), 465 375-385
36. Gentzel, I. N.; Ohlson, E. W.; Redinbaugh, M. G.; Wang, G. L. VIGE: virus-induced genome editing for improving abiotic and biotic stress 469 traits in plants. Stress Biology 2022, 2, 2.
38. Gong, Z.; Cheng, M.; Botella, J. R. Non-GM genome editing approaches in crops. Front. Genome Ed. 2021, 3, 81727
69. Zhang, X. H.; Tee, L. Y.; Wang, X. G.; Huang, Q. S.; Yang, S. H. Off-target effects in CRISPR/Cas9-mediated genome engineering. Mol. 533 Ther. Nucleic Acids 2015, 4, e264.
70. Liu, H.; Zhang, B. Virus-based CRISPR/Cas9 genome editing in plants. Trends Genet. 2020, 36, 810-813.
I understand that each author can present a different point of view, but how much further it can go?! This is not how review paper are supposed to be written and the presented manuscript is a review with too many review papers supporting it. Additionally, it is still very superficial in the main topics, where authors actually present some research papers.
Few formatting issues:
Point 3: “Geminivirus-based replicons enable highly efficient genome engineering”. It is in different/smaller font size than the rest of the manuscript.
Response 3: Thanks for pointing out that. We have corrected it in the revised manuscript.
Reviewer response to point 3: Thank you!
Point 4: There are “years” in references that are in bold, and some are not.
Response 4: We have corrected them in the revised manuscript.
Reviewer response to point 4: Thank you!
Related to the subject:
Point 5: “which allows the precise manipulation of plant genes”… Instead, of “gene” use “genome”.
Response 5: We have corrected it in the revised manuscript.
Reviewer response to point 5: Thank you!
Point 6: “however, this is limited by the rigid plant cell wall”… This is not the only limitation and depending on the methods used, all tissue culture challenges apply.
Response 6: Thanks for pointing out that. We revised the sentence to “however, this is limited by the tissue culture challenges”, that would be more rigorous.
Reviewer response to point 6: Thank you!
Point 7: “Here, we briefly describe CRISPR/Cas-based genome editing.” Currently, there are several good and recent reviews on this subject [1–4].
- Impens, L.; Jacobs, T.B.; Nelissen, H.; Inzé, D.; Pauwels, L. Mini-Review: Transgenerational CRISPR/Cas9 Gene Editing in Plants. Front. Genome Ed.2022, 4, 1–6, doi:10.3389/fgeed.2022.825042.
- Son, S.; Park, S.R. Challenges Facing CRISPR/Cas9-Based Genome Editing in Plants. Front. Plant Sci.2022, 13, doi:10.3389/fpls.2022.902413.
- Li, Y.; Li, W.; Li, J. The CRISPR/Cas9 Revolution Continues: From Base Editing to Prime Editing in Plant Science. J. Genet. Genomics2021, 48, 661–670, doi:10.1016/j.jgg.2021.05.001.
- Hassan, M.M.; Zhang, Y.; Yuan, G.; De, K.; Chen, J.-G.; Muchero, W.; Tuskan, G.A.; Qi, Y.; Yang, X. Construct Design for CRISPR/Cas-Based Genome Editing in Plants. Trends Plant Sci.2021, 26, 1133–1152, doi:10.1016/j.tplants.2021.06.015.
“We then introduce VIGE systems”. Currently, there are several reviews on this subject that also includes CRISPR/Cas-based genome editing. Additionally, “introducing” a subject will be first presented and this is not the case here. [5–7]
- Gentzel, I.N.; Ohlson, E.W.; Redinbaugh, M.G.; Wang, G.-L. VIGE: Virus-Induced Genome Editing for Improving Abiotic and Biotic Stress Traits in Plants. Stress Biol.2022, 2, 2, doi:10.1007/s44154-021-00026-x.
- Rössner, C.; Lotz, D.; Becker, A. VIGS Goes Viral: How VIGS Transforms Our Understanding of Plant Science. Annu. Rev. Plant Biol.2022, 73, 703–728, doi:10.1146/annurev-arplant-102820-020542.
- Oh, Y.; Kim, H.; Kim, S.-G. Virus-Induced Plant Genome Editing. Curr. Opin. Plant Biol.2021, 60, 101992, doi:10.1016/j.pbi.2020.101992.
Response 7: The reviewer is right. There are many reviews about CRISPR/Cas-based genome editing and VIGE, and we have cited reviews listed by the reviewer in the revised manuscript. Authors with different knowledge and professional backgrounds give readers different enlightenment. Here, we mainly focus on VIGE systems and the types of viruses used currently for CRISPR/Cas9 cassette delivery and genome editing. We summarize the advantages and disadvantages of utilizing these plant viral vectors, and the application of VIGE to a variety of host plants. We also discuss the potential bottlenecks and future prospects for VIGE in plants. Additionally, we corrected the sentence to “We then focus on VIGE systems” in the revised manuscript.
Reviewer response to point 7: Some review papers on the subjected were listed in first round of reviews by this reviewer. That was not a requested to add them in to the manuscript references. It was an example on how many already exist. Additionally, Liu, H.; Zhang, B. Virus-based CRISPR/Cas9 genome editing in plants. Trends Genet. 2020, 36, 810-813, recently summarized the advantages and disadvantages of utilizing these plant viral vectors in one of their topics “Viruses as Vectors for Delivering CRISPR/Cas9 Components”.
Point 8: The introduction is concise and clear, highlighting the main points. However, all other topics are very superficial in addressing the subjects and missing figures to better illustrate the importance and complexity of the theme. We must consider that several other strong and complete reviews on the subject have been recently published in important journals, such as the Annual Review of Plant Biology and Trends in Plant Science. The point here is that nowadays there is a flood of review papers, which requires the selection of the best quality ones. Additionally, this manuscript is very short, superficial, and does not present enough, and significant new information to be published in such an important journal as IJMS, but, with some improvement, this manuscript could suitable for publication in another and less impactful journal.
Response 8: Thanks for the suggestion. In the revised manuscript, we took the suggestion, enriched Part 3, highlighted significant works to better illustrate the importance and complexity of VIGE, added the comparison between the types of viral vectors and their advantages/disadvantages of use (Part 3.4), gave some perspective of this technology for crop breeding (Part 4), and cited review papers (Rössner et al. 2022, Annu. Rev. Plant Biol. 73, 703–728; Hassan et al. 2021, Trends Plant Sci. 26, 1133–1152) listed by the reviewer.
Reviewer response to point 8: I see the efforts of the authors to enrich and leave the surface of the subject, however, on trying to address the subject, more review papers were added to the review?! Figures had not changed, improved, or been added to help readers to better understand complexity of the theme.